# Design and Dynamic Analysis of a Novel Large-Scale Barge-Type Floating Offshore Wind Turbine with Aquaculture Cage

Yuting Zhai [1,2], Haisheng Zhao [1,2,*], Xin Li [1,2,*] and Wei Shi [1]

1   State Key Laboratory of Coastal and Offshore Engineering, Dalian University of Technology,
    Dalian 116024, China
2   School of Hydraulic Engineering, Faculty of Infrastructure Engineering, Dalian University of Technology,
    Dalian 116024, China
*   Correspondence: hzhao@dlut.edu.cn (H.Z.); lixin@dlut.edu.cn (X.L.)

**Abstract:** In this study, a novel large-scale barge-type floating offshore wind turbine with an aquaculture cage (LSBT-FOWT-AC) in a water depth of 100 m is designed through fully coupled analysis using the SESAM tool to support the Technical University of Denmark (DTU) 10 MW wind turbine. The intact stability and natural period of motion of the newly designed LSBT-FOWT-AC are evaluated based on the DNV rules and standards. Then, the dynamic responses of the LSBT-FOWT-AC under various sea conditions are studied. The motion of the LSBT-FOWT-AC platform is considerably affected by waves, and its motion response is within a reasonable range even under the extreme sea conditions of the 100-year return period. By analyzing the results of the out-of-plane bending moment of root of blade 1 (RootMyc1), it can be seen that the rotor frequency (1P) has a visible influence on the wind turbine. Through the analysis of dynamic response statistics of the LSBT-FOWT-AC structure by the single variable method of environmental loads, it is found that wind force exerts the greatest impact on the dynamic response compared to the wave-excitation force and current drag force.

**Keywords:** offshore wind turbine; aquaculture cage; barge-type; motion response; mooring system

## 1. Introduction

Floating offshore wind turbines have been proven as one of the main elements of future green transition in the world. Many countries have begun to vigorously develop offshore wind power green energy to replace fossil energy. Figure 1 shows the cumulative installed capacity of offshore wind power in the world in the past ten years. It can be seen that there has been a significant increase in the installed capacity of offshore wind turbines in 2021. However, current wind energy development is mainly located in shallow water areas, where the available space and energy are limited. Additionally, with the continuous development of human being populations, pollution in shallow water will inevitably occur. In order to alleviate these problems, developing wind power in deep sea has begun to be addressed.

Compared to shallow water, the deep sea has more space and resources, but it is accompanied by more complex environmental loads and construction conditions. For wind turbines in shallow water, their foundation form is mostly fixed, but the high cost of the fixed type makes it no longer applicable in the deep sea [1], which indicates that it needs to be replaced by a floating platform form. Compared to the fixed-type wind turbine, the floating offshore wind turbine (FOWT) is a higher dynamic system because it is simultaneously affected by wind, current and wave loads and constrained by the mooring system [2]. At present, the development of global floating offshore wind turbines is in the stage of technological breakthroughs. In 2021, China's first FOWT was installed and successfully connected to the grid for power generation in the Yangjiang River, Guangdong Province.

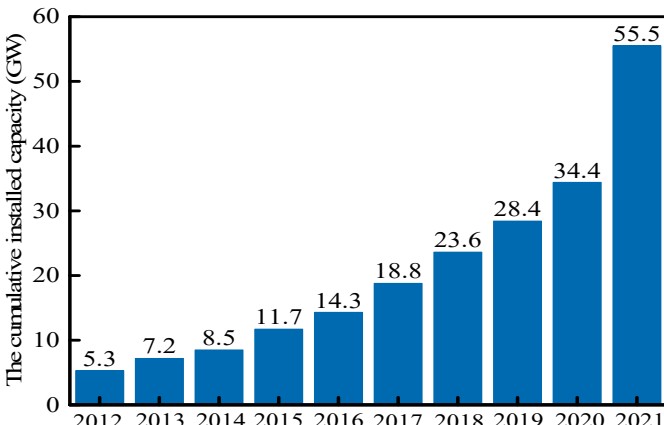

**Figure 1.** The cumulative installed capacity of offshore wind power in the world in the past ten years.

With regard to floating offshore wind turbines, there are four main types: spar type [3], semi-submersible type [4], tension leg type [5] and barge type [6]. Ye and Ji [7] studied the dynamic response of a spar-type direct-drive wind turbine subjected to external and internal excitations. Soeb et al. [8] analyzed the nonlinear motion response of a spar platform under wave and current loads. Based on the viscous flow theory, Tran and Kim [9] carried out a numerical study on the hydrodynamic characteristics of a semi-submersible FOWT under waves. Bayati et al. [10] studied the dynamic response characteristics of semi-submersible platforms under different water depths. Oguz et al. [11] studied the Iberdrola TLP FOWT experimentally and numerically under realistic wind and wave conditions. They found that the effect of wind significantly contributed to the overall response of the platform, while changes in wave conditions had relatively little effect on the platform response. So far, most of the studies have been focused on the spar type, semi-submersible type and tension leg type. This is because the hydrodynamic performance of the barge platform was not satisfactory enough compared to other platforms in the past; in recent years, however, IDEOL [12] developed the concept of "moon pool" (a pool that runs up and down in the middle of the barge platform) to improve the hydrodynamic performance of barge-type platforms, reduce the kinematic response and construction costs. Now, the barge-type platform is also starting to be noticed.

Ikoma et al. [6] studied the motion responses of a barge-type floating system with four moon pools and a vertical-axis wind turbine by experiment and numerical simulation. Through this study, it was found that the water motions and free-surface shapes were different at particular frequencies in the front and rear moon pools. Yang et al. [13] analyzed the mooring tension of the coupled tunnel–barge system in waves, where the restraining effect of the steel four-leg catenary mooring system on the tunnel–barge system and the changing law of tension were considered in detail. The research of Chuang et al. [14] found that the stability of the barge-type platform should be improved in the pitch rotation in extreme conditions. Yang et al. [15] investigated the impact of a mooring breakage on the dynamic responses of the rotor, platform and remaining mooring lines of the barge-type FOWT, and demonstrated the benefit of shutdown in ensuring the safe operation of the FOWT with mooring line breakage. A preliminary study of the Japan Kitakyushu barge-type FOWT demonstration project was conducted by Kosashi et al. [16], where the response amplitude operator, statistical value and amplitude spectral densities obtained from experimental tests were compared with the simulation results, and a good agreement was observed.

The above work on the barge-type FOWT is mainly focused on small or medium-sized wind turbines with 5 MW and below. With the vigorous development of green energy, 5 MW FOWTs are no longer enough to meet the demand, and large-scale offshore wind power has begun to appear due to the benefits of more power per unit of sea area and better economics of scale. A 10 MW reference wind turbine [17] was proposed by the Technical

University of Denmark (DTU) Wind Energy and Vestas. The 10 MW FOWT, with larger turbine blades, tower and floating platform, inevitably undertakes larger local loads and overturning moments induced by wind, wave and current [18]. These features significantly increase platform heeling motion, unstable aerodynamics, structural vibrations, fatigue and extreme loads on the tower and blades. Therefore, the dynamic response and operating safety of the 10 MW FOWT under various sea conditions are required to be considered in detail. Zhao et al. [19] presented a conceptual semi-submersible platform to support a 10 MW wind turbine through aero-hydro-servo-elastic fully coupled analysis using the OpenFast v2.4.0 code. Based on the structure designed by DNV, the complete stability and dynamic response of the newly designed semi-submersible FOWT under different fault conditions were studied. An optimal semi-submersible platform was designed by Ferri et al. [20] to support the 10 MW wind turbine by adjusting the diameters of the outer cylinders and radial distances from the center of the platform. Ahn and Shin [21] performed the model testing and numerical simulation of a 10 MW FOWT to reveal its dynamic characteristics induced by regular waves over different periods. At present, there have been some studies on semi-submersible 10 MW FOWTs, while the related research on 10 MW barge-type FOWTs are still scarce.

Currently, deep-sea aquaculture has become another new focus of development in various countries due to its high return. In order to improve the technology and benefits of deep-sea aquaculture, some new forms of aquaculture cages have been proposed by combining technologies from fish farming and offshore infrastructures, such as the vessel-shaped "Havfarm" concept, the "Arctic Offshore Farming" concept, the "egg" closed cage concept and the semi-submersible "Sea Farm 1" concept [22]. Considerable studies have been carried out to investigate the characteristics of deep-sea aquaculture cages. Zhao et al. [23] considered the effects of plane net inclination angles, heights, distance between two nets and nets number on the flow field around the plane net by physical model test and numerical simulation. Bi et al. [24] obtained the flow field of the planar net with different degrees of biological contamination by submerging it in different water depths and for different durations; then, the empirical formulas of predicting the resistance and downstream flow velocity of waterborne fouling networks were proposed by experiments and numerical simulations. Dong et al. [25] conducted a series of experiments on different types of net planes in the wave trough and developed a net plane wave force model that can accurately calculate the wave force on the net plane. By comparing the numerical and experimental results, a method for the structural analysis of aquaculture nets was developed and it was concluded that the resistance load and cage volume depend on the size and weight of the net system [26].

In recent years, the combination concept of the aquaculture cage and the FOWT has been proposed, which can make use of the sea space more efficiently and lower the construction cost by sharing the floating platform. Moreover, the electricity generated by the FOWT can also be directly supplied to the aquaculture equipment. Based on the above advantages, this concept attracted the attention of scholars as soon as it was proposed. Lei et al. [27] designed a floating offshore wind turbine with a steel fish cage (FOWT–SFFC), and studied the motion response and nonlinear dynamic performance of the structure. Liang et al. [28] developed an offshore floating multipurpose platform for Blue Growth Farm that supports a 10 MW wind turbine on one side of the platform with fish cages, and the results of numerical simulation show that the hydrodynamic characteristics of the multifunctional platform are excellent. Abhinav et al. [29] proposed a novel multi-purpose platform (MPP) by retrofitting a feed barge with a small wind turbine and analyzing its performance in the frequency domain. Zhai et al. [30] used AQWA to investigate the hydrodynamics of a barge-type FOWT with an aquaculture cage and found that the presence of the aquaculture cage makes the structure more sensitive to current.

In summary, the related conceptual designs and dynamic analyses on the integrated structure of combing an FOWT and aquaculture cage are relatively few in comparison to the FOWT or aquaculture cage, and much lesser in comparison to the large-scale barge-type

FOWT with an aquaculture cage. However, relative to other types of floating platform, the barge-type platform is simple in structure, low in cost and long in life; moreover, a moon pool throughout the barge-type platform provides additional space for mariculture and the fish farming equipment can be placed on the platform directly. Therefore, research on the large-scale barge-type floating offshore wind turbine with an aquaculture cage (LSBT-FOWT-AC) is necessary for the development of deep-sea energy.

In this study, a novel LSBT-FOWT-AC is designed and its dynamic responses under various sea conditions are analyzed. The remainders of this paper are organized as follows: Section 2 introduces the theoretical methods used in this research; Section 3 introduces the design process, stability analysis and simulation environment of the LSBT-FOWT-AC. In Section 4, the results and discussion of this study are reported. Section 5 presents the conclusions of this work. Finally, Section 6 presents the future work.

## 2. Theoretical Methods

The LSBT-FOWT-AC is a multi-body coupled system affected by various environmental loads, and its dynamic response is very complex. Therefore, the fully coupled numerical simulation of the LSBT-FOWT-AC involves multi-disciplinary theory such as aerodynamics, hydrodynamics, structural dynamics, multibody dynamics [31] and turbine control.

### 2.1. Equation of Motion in the Time Domain

The 6-DOF time-domain rigid body motion control equation [32] of the floating platform in the LSBT-FOWT-AC can be written as:

$$(\boldsymbol{M} + \boldsymbol{A}_\infty)\ddot{\boldsymbol{x}}(t) + \int_0^t \boldsymbol{K}(t - \tau)\dot{\boldsymbol{x}}(\tau)d\tau + \boldsymbol{C}\boldsymbol{x}(t) = \boldsymbol{F}_{wind}(t) + \boldsymbol{F}_{wave}(t) + \boldsymbol{F}_{curr}(t) + \boldsymbol{F}_{moor}(t) \quad (1)$$

where $\boldsymbol{x}$, $\dot{\boldsymbol{x}}$ and $\ddot{\boldsymbol{x}}$ represent the six degrees of freedom (DOF) displacement, velocity and acceleration of the LSBT-FOWT-AC platform, respectively; $\boldsymbol{M}$ is the mass matrix; $\boldsymbol{A}_\infty$ is the infinite-frequency added mass matrix; $\boldsymbol{K}$ represents the wave–radiation–retardation kernel matrix; $\boldsymbol{C}$ is defined as hydrostatic stiffness matrix; $\boldsymbol{F}_{wind}$ is the wind loads on the LSBT-FOWT-AC obtained by the wind speed-thrust curve; $\boldsymbol{F}_{wave}$ and $\boldsymbol{F}_{cuur}$ are the wave excitation load and current force, respectively, which are calculated by potential flow theory and Morison equation; $\boldsymbol{F}_{moor}$ represents the restoring force of the mooring system.

$$\boldsymbol{K}(t) = \frac{2}{\pi} \int_0^\infty \boldsymbol{b}(\omega) \cos(\omega t)d\omega \quad (2)$$

where $\boldsymbol{b}$ is the linear radiation damping matrix.

### 2.2. Aerodynamic Loads

The blade element momentum (BEM) theory [33,34] assumes that the blade is divided into small elements of infinitely thin radial thickness, called "blade elements". When the wind flows through the blade element, the pressure difference on the surface of the blade element forms the momentum change of the airflow in and out of the blade element, thereby promoting the rotation of the blades to function. Based on the semi-empirical Beddoes–Leishman dynamic stall model [35], unsteady aerodynamics are considered while ignoring the aerodynamic effects of tip losses, hub losses, oblique wake and tower shadow effects [36].

Figure 2 illustrates the velocity component and force diagrams of the blade element, where $U_\infty$ represents the wind speed at infinity; $W$ is the relative wind speed; $r$ is the distance between the blade element and the center of the hub; $C$ is the chord length of the blade; $b$ and $b'$ represent the axial and tangential induction factors; $\Omega$ is the rotational angular velocity; $\varphi$ is the inflow angle; $\beta$ is the pitch angle; $\alpha$ is the airfoil angle of attack; $dD$ and $dL$ are the drag and lift of the blade element; $dF_x$ and $dF_y$ are the forces along the X and Y directions of the blade element, respectively.

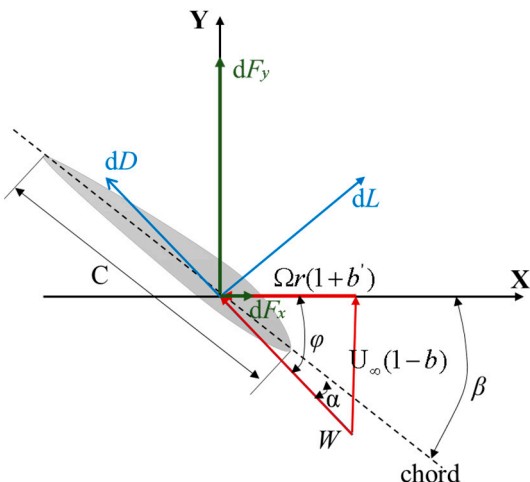

**Figure 2.** The velocity component diagram and force diagram of blade element.

The relative wind speed of the blade element can be written as:

$$W = \sqrt{U_\infty^2(1-b)^2 + r^2\Omega^2(1+b')^2} \tag{3}$$

Therefore, the thrust and torque experienced by the blade element at the blade radius $r$ are:

$$dT_x = NdF_x = \frac{1}{2}\rho NCW^2(C_L\cos\varphi + C_D\sin\varphi)dr \tag{4}$$

$$dM_x = NrdF_y = \frac{1}{2}\rho NrCW^2(C_L\sin\varphi - C_D\cos\varphi)dr \tag{5}$$

where $N$ is the number of blades; $C_D$ and $C_L$ are the drag and lift coefficients of the blade element, respectively; $\rho$ is the density of air, taking 1.29 kg/m$^3$.

By integrating $dT_x$ and $dM_x$ along the blade span, the thrust and torque of the rotor can be obtained.

### 2.3. Potential Flow Theory [37]

The existence of large marine engineering structures such as a floating platform will change the wave field around it. When calculating the wave force, the influence of the diffraction and radiation effect must be considered, so the potential flow theory needs to be used.

The wave-excitation forces of the floating platform can be written as a series of expansions,

$$\boldsymbol{F} = \boldsymbol{F}^{(1)} + \boldsymbol{F}^{(2)} + \ldots + \boldsymbol{F}^{(n)} + \ldots \tag{6}$$

where $\boldsymbol{F}^{(1)}$, $\boldsymbol{F}^{(2)}$ and $\boldsymbol{F}^{(n)}$ are the first-order, second-order and $n^{th}$-order velocity potentials, respectively.

In this paper, only first-order wave-excitation forces are used. Under random waves, the first-order wave-excitation force received can be calculated by the following formula:

$$\boldsymbol{F}^{(1)}(t) = \text{Re}\sum_j^N A_j\boldsymbol{X}^{(1)}(\omega_j, \beta_j)e^{i\omega_j t} \tag{7}$$

where $A_j$ is the magnitude of a regular incident wave of frequency $\omega_j$ and direction $\beta_j$; $\boldsymbol{X}^{(1)}$ is the response amplitude operator (RAO).

### 2.4. Morison Equation

Small-scale cylindrical structures such as aquaculture cage frames have no evident effect on wave field, and the wave excitation force can be calculated using the Morison

equation [38]. According to the Morison equation [25], the wave-excitation force can be divided into two parts: drag force and inertia force, which are written as follows:

$$F(t) = \frac{1}{2}\rho C_d Du(t)|u(t)| + (1 + C_m)\rho\frac{\pi D^2}{4}\dot{u}(t) \tag{8}$$

where $u(t)$ and $\dot{u}(t)$ are the horizontal velocity and acceleration in the wave's water particles; $C_d$ is the drag coefficient and $C_m$ is the additional mass coefficient.

## 3. Numerical Model and Model Testing

### 3.1. Design of the LSBT-FOWT-AC

An important objective of this work is to design a large-scale barge-type floating offshore wind turbine with an aquaculture cage (LSBT-FOWT-AC) whose platform is used to support a DTU 10 MW wind turbine in a water depth of 100 m. Fully coupled aero-hydro-servo time-domain simulations are conducted for the 10 MW DUT LSBT-FOWT-AC using the analysis tool SESAM [39]. As shown in Figure 3, the main dimensions and general arrangement of the LSBT-FOWT-AC are first determined using GeniE according to the design requirements, including water depth, kinetic performances of the platform and the wind turbine, estimated structural weight, and so on. Then, the stability analysis and preliminary frequency hydrodynamic analysis under the condition of no mooring system are carried out by the analysis tool HydroD on the preliminarily determined platform with the aquaculture cage, so as to judge whether the main dimensions of the integrated structure meet the design requirements. Finally, the mooring system is designed by MIMOSA [40] according to the environmental parameters for the LSBT-FOWT-AC that meets the design requirements. After the design of the LSBT-FOWT-AC is completed, the frequency domain hydrodynamic results of the platform are imported into the analysis tool SIMA [41] for establishing the overall coupling model and performing time-domain calculations.

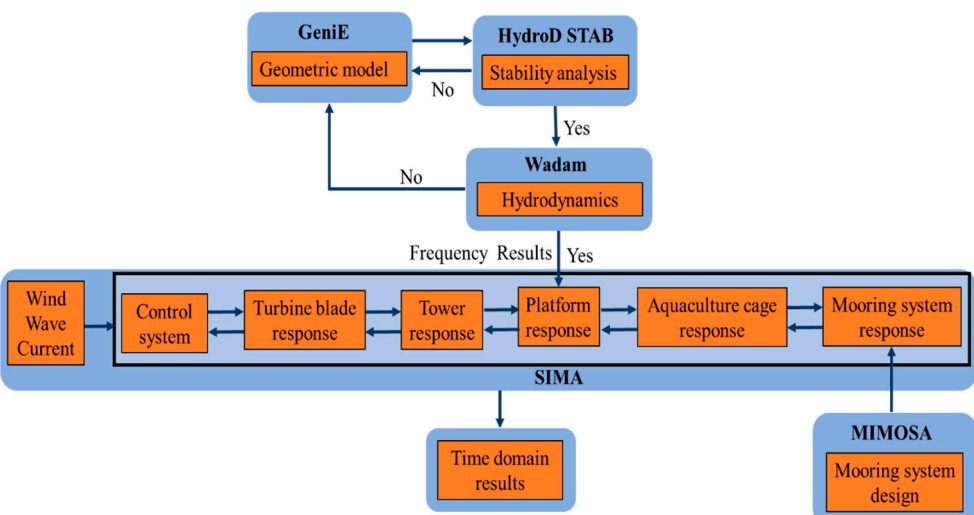

**Figure 3.** Concept design process of the LSBT-FOWT-AC.

The LSBT-FOWT-AC's structure is mainly composed of four parts: wind turbine, platform, aquaculture cage and mooring system. The couplings among environmental loads, control systems and structural dynamics are handled by SIMA Simo-Riflex-AeroDyn (SRA), where the structural hydrodynamics is considered through Simo and Riflex, whereas the aerodynamics of the turbine blades is carried out through AeroDyn. It is noted that the LSBT-FOWT-AC's structure is a more complex integrated system compared to fixed offshore wind turbines. The properties of the DTU 10 MW wind turbine [17] are listed in Table 1.

**Table 1.** Properties of DTU 10 MW offshore wind turbine.

| Parameters | Value |
| --- | --- |
| Wind Regime | IEC class 1 A |
| Cut-in wind speed (m/s) | 4.0 |
| Rated wind speed (m/s) | 11.4 |
| Cut-out wind speed (m/s) | 25.0 |
| Rotor diameter (m) | 178.3 |
| Hub height (m) | 119.0 |
| Shaft tilt angle (deg) | 5.0 |
| Accumulated mass of three blades (kg) | 230,667 |
| Nacelle mass (kg) | 446,036 |
| Tower mass (kg) | 591,758 |
| Rotor frequency (1P, Hz) | 0.1–0.16 |
| Natural frequency of 1st tower mode (Hz) | 0.25 |

The illustration of the concept of the LSBT-FOWT-AC is shown in Figure 4, and Table 2 gives the detailed parameters of the barge platform. It can be seen that the main sizes of the platform adopt the form of 80 m × 60 m × 17 m to better support the 10 MW wind turbine. The barge-type platform contains two identical moon pools with lengths, widths and depths of 30 m, 20 m and 17 m, respectively, and the space between one another is 10 m. In order to facilitate the maintenance of the wind turbine, the wind turbine is arranged on the long side of the platform, and the center of gravity of the platform in the x direction is adjusted to 0.574 m by ballast for the overall stability. The aquaculture cage is located directly below the barge-type platform, and Table 3 presents the detailed parameters of the aquaculture cage.

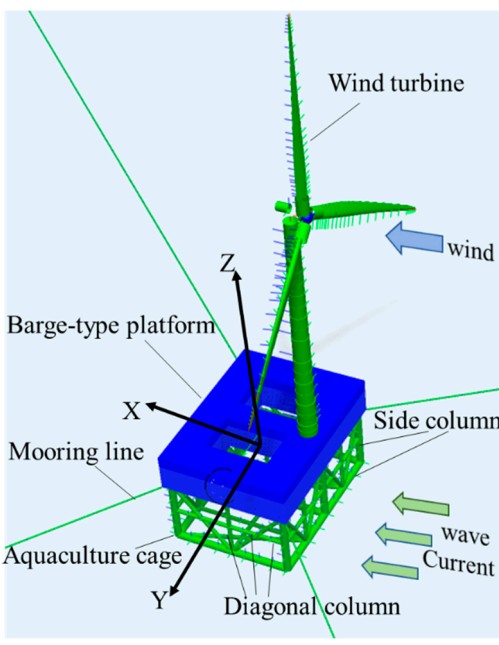

**Figure 4.** Illustration of the concept of LSBT-FOWT-AC.

**Table 2.** Main parameters of the barge-type platform.

| Parameters | Value |
|---|---|
| Barge size (W × L × H) (m) | 80 × 60 × 17 |
| Moon pool (W × L × H) (m) | 20 × 30 × 17 |
| Draft (m) | 12 |
| Mass (kg) | 8,471,280 |
| COG (m) | (0.574, 0, −5.18) |
| Roll inertia (kg/m$^3$) | $1.40 \times 10^{10}$ |
| Pitch inertia (kg/m$^3$) | $2.55 \times 10^{10}$ |
| Yaw inertia (kg/m$^3$) | $3.59 \times 10^{10}$ |

**Table 3.** Main parameters of the aquaculture cage.

| Parameters | Value |
|---|---|
| Diameter of side column and diagonal column (m) | 3.5 |
| Diameter of diagonal column (m) | 2 |
| Thickness of the side column and diagonal column (m) | 0.02 |
| Height (m) | 30 |

### 3.2. Stability Analysis

Since the novel LSBT-FOWT-AC is subjected to non-negligible wind loads in the operating state, the wind heeling moment needs to be considered in the stability test, which is set as the maximum moment at the rated wind speed of the wind turbine for the sake of conservation and its value is calculated as 184,073 kN·m. There are two requirements regarding the intact stability of the barge-type platform FOWT considered [42]: first, the area under the righting moment curve should be equal to or greater than 1.4 times the area under the wind heeling moment curve; second, the righting moment shall be positive over the entire range of angles from upright to the second intercept. Figure 5 shows the wind heeling moment and righting moment of the integrated structure when the waves are incident along the x direction. It can be seen that the design of the LSBT-FOWT-AC meets the stability requirements.

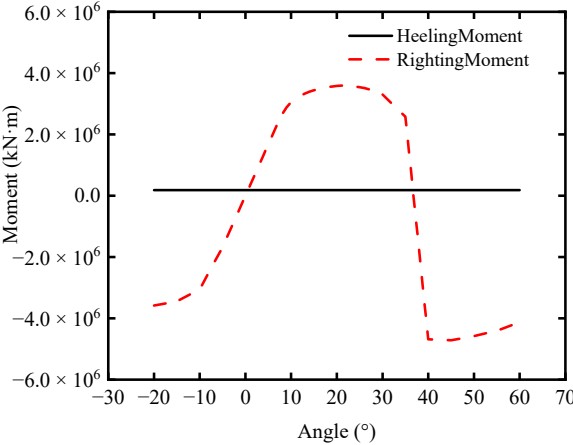

**Figure 5.** The wind heeling moment and righting moment of LSBT-FOWT-AC.

### 3.3. Mooring System Analysis

Based on the integrated structure mentioned above, the mooring system is designed under the environmental conditions of the East China Sea at a water depth of 100 m. The LSBT-FOWT-AC can be subjected to environmental loads in any direction during operation,

so the mooring system adopts a symmetrical distribution. Four mooring lines in the form of catenary lines with a length of 620 m and an included angle of 90° are arranged at each right angle of the platform at a water depth of 12 m, and the length of its lying section is 372 m. The main parameters of the mooring system can be found in Table 4. Additionally, the schematic layout of the mooring lines is illustrated in Figure 6.

**Table 4.** Properties of the mooring system.

| Parameters | Value |
| --- | --- |
| Unstretched line length (m) | 620 |
| Diameter of each mooring line (m) | 0.153 |
| Line mass density in water (kg/m) | 401 |
| Line type | R4-Studless |
| External area of each mooring line (m$^2$) | 0.059 |
| Elastic stiffness, EA (kN) | $3.7789 \times 10^6$ |
| Breaking loads (kN) | 20,156 |

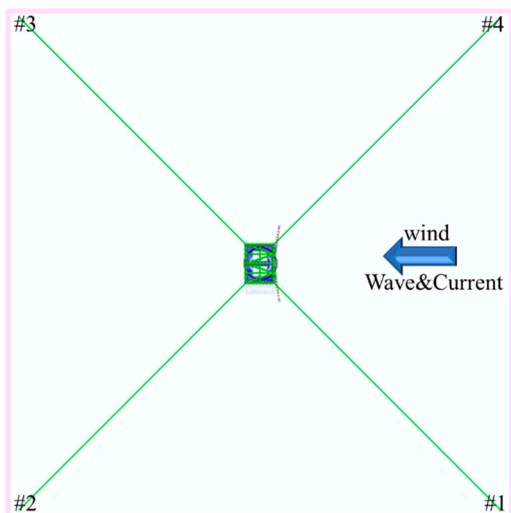

**Figure 6.** The schematic layout of mooring lines.

*3.4. Environmental Conditions*

In this study, the environmental conditions (ECs) with irregular waves, steady winds and currents are selected to investigate the hydrodynamic performance of the LSBT-FOWT-AC, and the JONSWAP spectrum is used to generate the irregular wave time series. Ten typical environmental conditions are defined in Table 5, including different wind speeds, wave heights and current speeds, according to real sea environmental conditions of the East China Sea. As shown in Figure 4, the wind, current and wave directions are all along the positive x-axis in the numerical calculation. Each simulation lasts 4200 s and the results from the first 600 s are eliminated to mitigate the start-up transient effect.

**Table 5.** Environmental conditions.

| | Wind Speed (m/s) | $H_s$ (m) | $T_p$ (m) | Current Speed (m/s) | Turbine Status |
|---|---|---|---|---|---|
| EC1 | 6 | 2.50 | 6 | 0.81 | Operating |
| EC2 | 11.4 | 3.52 | 7.9 | 0.81 | Operating |
| EC3 | 16 | 6.85 | 9.4 | 0.81 | Operating |
| EC4 | 40 | 14 | 14.5 | 0.81 | Shutdown |
| EC5 | 11.4 | 5.55 | 8.8 | 0.81 | Operating |
| EC6 | 11.4 | 6.85 | 9.4 | 0.81 | Operating |
| EC7 | 6 | 3.52 | 7.9 | 0.81 | Operating |
| EC8 | 16 | 3.52 | 7.9 | 0.81 | Operating |
| EC9 | 11.4 | 3.52 | 7.9 | 0 | Operating |
| EC10 | 11.4 | 3.52 | 7.9 | 2 | Operating |

## 4. Results and Discussion

### 4.1. Free-Decay Simulation

The simulation tests for free-decay in six degrees of freedom (DOF) are carried out in SIMA by applying an excitation (force or moment) on the center of gravity of the LSBT-FOWT-AC for 10 s. The excitation will provide an initial offset at the corresponding DOF. Then, this excitation is removed in the simulation so that the LSBT-FOWT-AC starts oscillating freely and decaying. Figure 7 presents the time histories of the surge free-decay test. The free-decay tests process of the remaining five DOF are the same with the case of surge, and Table 6 lists the natural periods of the LSBT-FOWT-AC. It can be observed that the natural periods of the LSBT-FOWT-AC are in accordance with the range recommended by specification DNV-RP-0286 [43].

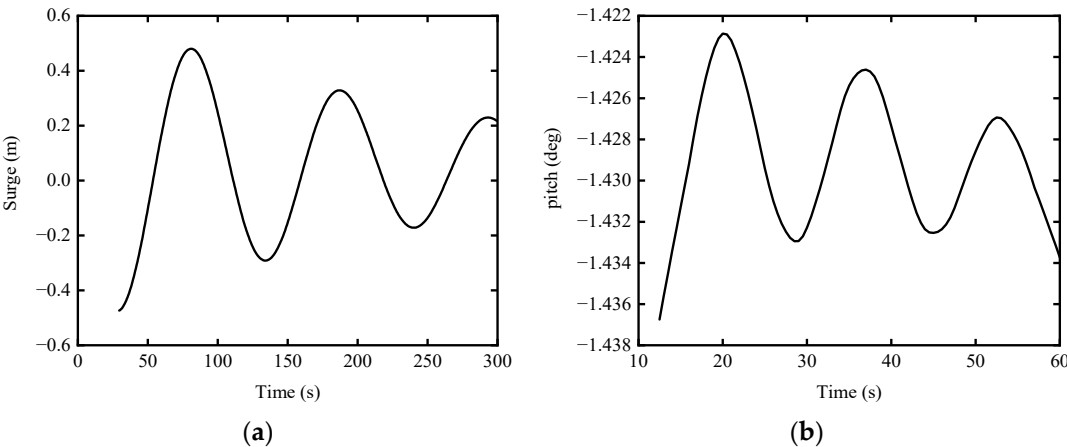

**Figure 7.** Platform response time series of surge and pitch in decay test: (**a**) Surge; (**b**) Pitch.

**Table 6.** The natural periods and natural frequencies of the platform response.

| Motion | Natural Period | Natural Frequency |
|---|---|---|
| Surge | 106 s | 0.0094 Hz |
| Heave | 10 s | 0.1 Hz |
| Pitch | 16.5 s | 0.0606 Hz |

### 4.2. Analysis of Motion Responses of Integrated Structure

Figure 8 shows the time histories of the mooring line tension of the LSBT-FOWT-AC at EC2. It can be seen that the upstream suspension #1 and #4 are in a tight state, and the mooring rope tension is large; the downstream suspension #2 and #3 are in a loose state, and the tension of the mooring line is small, so suspension #1 is selected to study the behavior of anchor chain tension at different environmental conditions.

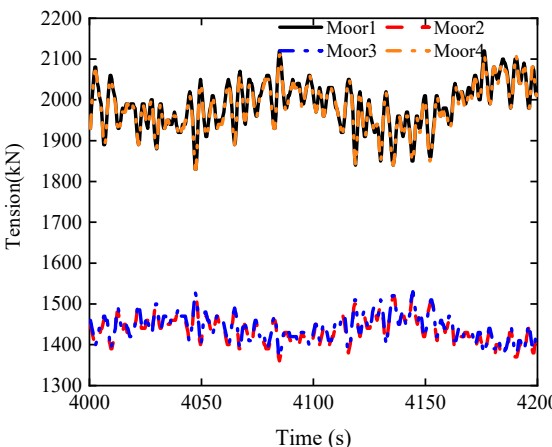

**Figure 8.** Time histories of mooring lines tension at EC2.

EC1–EC4 in Table 5 are selected to study the mooring lines tension and motion response of the LSBT-FOWT-AC in actual operation. Figure 9 shows the time histories and power spectral density (PSD) spectra results of the #1 tension under these four conditions. It can be seen from Figure 9a that the tension of the mooring line #1 fluctuates around 2000 kN. The greater the wind speed and wave height, the greater the fluctuation amplitude of the tension. For the case of EC3 with the maximum wind speed and wave height in the operating states of Table 5, the maximum tension of the mooring line #1 is only 2200 kN, and the maximum tension can reach about 9000 kN in extreme sea conditions (EC4), whereas all the cases are less than the breaking tension 20,156 kN, indicating the reliability of the mooring system in the LSBT-FOWT-AC.

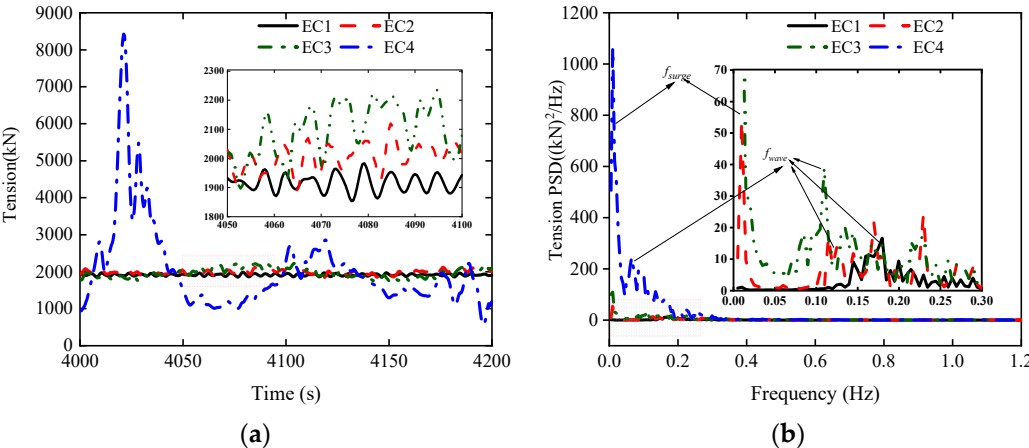

**Figure 9.** Time histories and spectra of the #1 tension under real environmental conditions: (**a**) Time histories of the #1 tension; (**b**) Spectra of the #1 tension.

Figure 9b presents the PSD spectra results of the mooring line #1 tension, which are dominated by the wave frequency and nature frequency of platform surge under these four sea conditions. Furthermore, it is also observed that the variation law of tension in the time domain is similar to that of platform surge because the movement of platform surge does affect the change of mooring lines tension.

For floating offshore wind turbines, the motion response of the platform is an important parameter for characterizing the structure performance, which also applies to the LSBT-FOWT-AC. Figure 10 shows the time histories and spectra results of motion responses of the platform surge, heave and pitch in the LSBT-FOWT-AC.

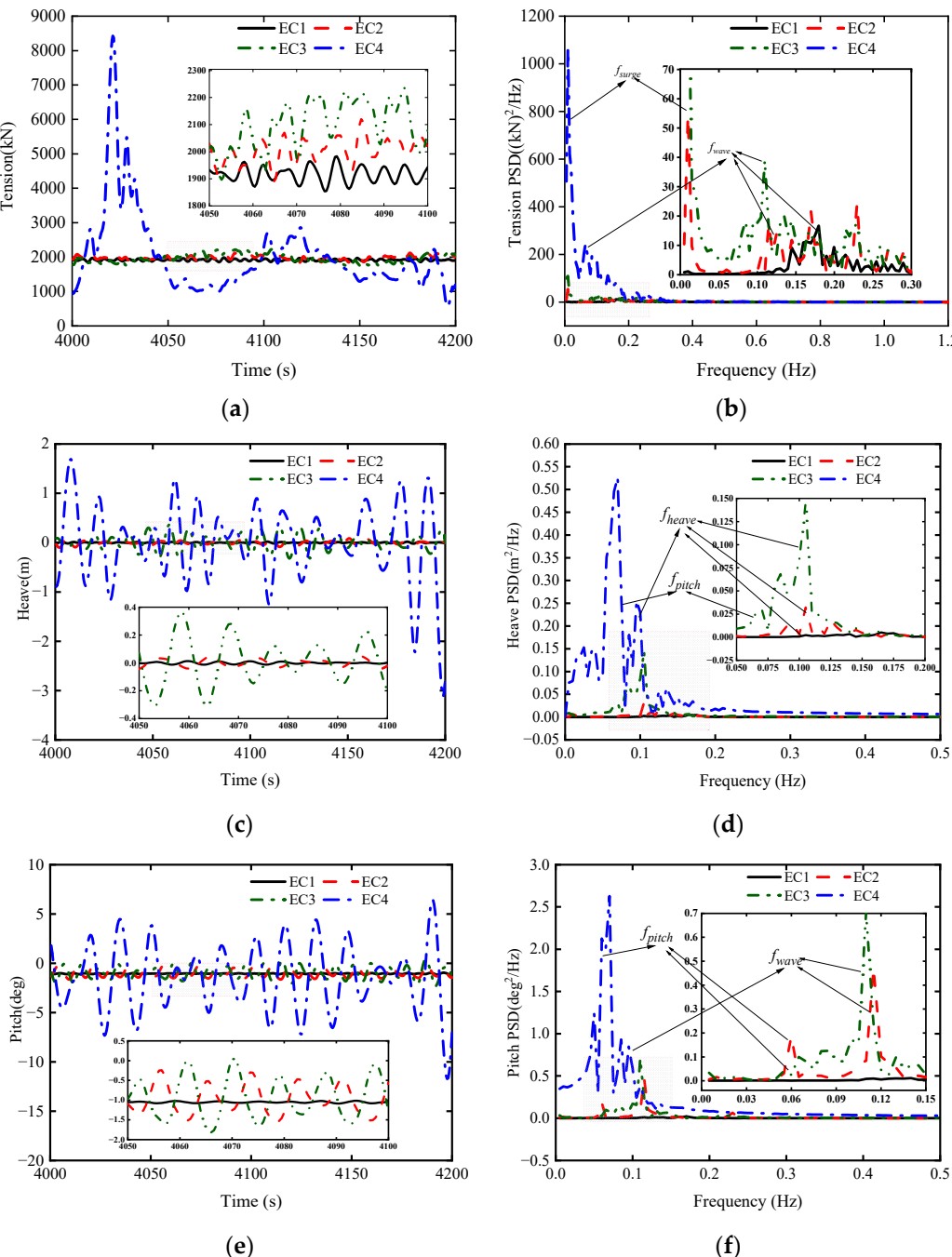

**Figure 10.** Time histories and spectra of the platform surge, heave and pitch motions: (**a**) Time histories of the surge; (**b**) Spectra of the surge; (**c**) Time histories of the heave; (**d**) Spectra of the heave; (**e**) Time histories of the pitch; (**f**) Spectra of the pitch.

Figure 10a,b presents the results of surge motion of the platform. Similar to the #1 tension, it can be found that the platform surge fluctuates around a value of about 2.6 m, and with the increase in wind speed and wave height, the fluctuation amplitude becomes larger. Under normal operating sea conditions, the maximum value of surge is less than 5 m. Therefore, the structure operates stably under normal sea conditions and meets the requirements of surge design of the floating structure. For extreme sea conditions with the return period of 100 years, viz., EC4, the maximum surge is less than 15 m most of the time, but there are occasional dangers when reaching 20 m. Moreover, it is noted from the spectra analysis that the surge motion of the platform is dominated by the wave frequency and nature frequency of platform surge, similar to that of the mooring line #1 tension.

Figure 10c,d presents the results of the heave motion of the platform. The fluctuation range of the platform heave is very small. Under normal sea conditions, namely EC1–EC3, the maximum value of the heave is generally not more than 0.4 m, and even in the EC4 sea state, the maximum value of the heave is only 3 m. This shows that the stability of the LSBT-FOWT-AC in the vertical movement is excellent, and this performance is important for fish farming. In addition to the nature frequency of the heave, the frequency component of the pitch rotation is also observed in Figure 10d with a distinct peak at about 0.0606 Hz, indicating a coupling effect between the heave and pitch motions.

Figure 10e,f presents the results of pitch rotation of the platform, and excellent stability performance is observed in the pitch rotation as well. Under the EC1–EC3 operating conditions, the maximum value of pitch rotation does not exceed 2°, and the corresponding value does not exceed 15° in extreme sea conditions. The designed LSBT-FOWT-AC satisfies the relevant preliminary requirements recommended by DNV [29]. It is found from the spectra analysis that the platform pitch rotation is dominated by the wave frequency and its nature frequency. In summary, the wave frequency has a great influence on the surge and pitch motion responses; thus, the natural frequency of the surge and pitch must be avoided in the wave-frequency range in the design process.

Determining the structural loads at key components, such as the blade root moment, is crucial to the overall performance and operational stability of the integrated LSBT-FOWT-AC system. Figure 11 shows the time histories and spectra results of the out-of-plane bending moment of root of blade 1 (RootMyc1) in the LSBT-FOWT-AC. It can be seen that when the wind turbine is in the operating state, the RootMyc1 increases with the increase in wind speed, and the RootMyc1 is almost zero when the wind turbine is parked. When the wind speed is between the rated wind speed and the cut-in wind speed, the order of magnitude for the RootMyc1 is $10^4$. When the wind speed exceeds the rated wind speed, the value of the RootMyc1 will increase rapidly. For example, the order of magnitude reaches $10^5$ when the wind speed is 16 m/s. Furthermore, it can be seen from Table 1 that the RootMyc1 has peaked values around the wave frequency, rotor frequency (1P), three times rotor frequency (3P) and natural frequency of the first tower mode. Therefore, avoiding the rotor frequency of 1P in the design of integrated structure is of great importance for the normal operation of the wind turbine.

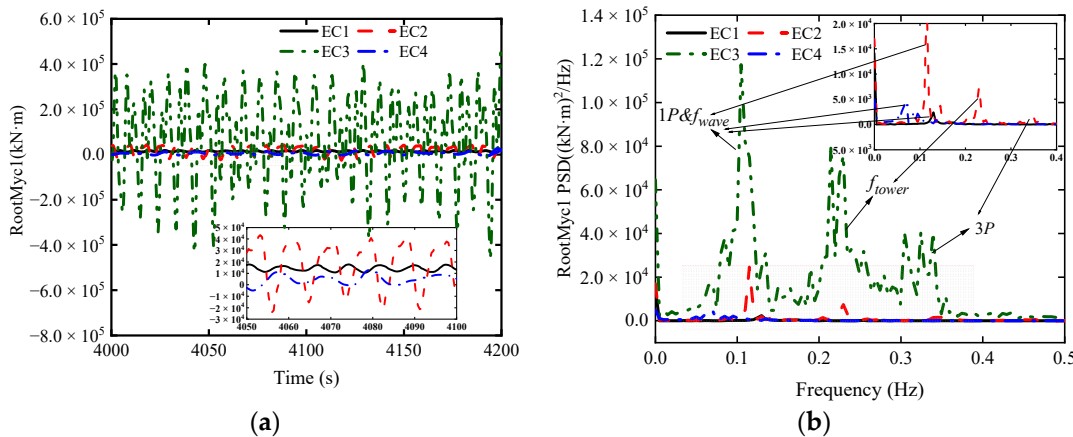

**Figure 11.** Time histories and spectra of the RootMyc1 under real environmental conditions: (**a**) Time histories of the RootMyc1; (**b**) Spectra of the RootMyc1.

### 4.3. Statistical Analysis of Motion Responses of Integrated Structure

In order to further study the influence of wind speed, wave height and current speed on the LSBT-FOWT-AC structure, EC5–10 are set up. Compared with the rated case EC2, these six environment conditions only have one environmental variable, wherein EC2, EC5 and EC6 have different wave heights; EC7, EC2 and EC8 have different wind speeds; and EC9, EC2 and EC 10 have different current speeds.

Figure 12 presents the statistics of the #1 tension involving mean, standard deviation (Std), maximum (Max) and minimum (Min). It can be seen that the mean value of the #1 tension hardly changes with the variation of wave height, whereas the Std, maximum and minimum values all increase with the increase in wave height. Therefore, the increase in wave height mainly affects the fluctuation amplitude of the #1 tension. However, all four statistical values change with the wind speed, which is consistent with the changing law of the thrust on the wind turbine. Moreover, the mean value of the #1 tension increases slightly with the increase in current speed. For example, the mean value of the #1 tension only increases by about 0.5% when the current speed increases by about 1 m/s. The effect of current speed on the Std is also small, resulting in similar differences for these four statistical values under different sea conditions (EC2, EC9 and EC10). In general, the effect of current speed on the motion responses of the LSBT-FOWT-AC structure is limited compared to those of wind speed and wave height.

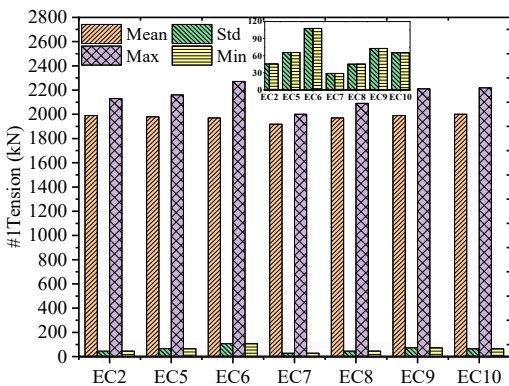

**Figure 12.** Statistics of the #1 tension.

Figure 13 shows the statistics of platform motion response. For the platform surge motion, it can be seen that its change rule is the same as that of the #1 tension, but the change range is more evident. This is because the change in the #1 tension is affected by the surge motion. The apparent change in the magnitude of surge compared to the #1 tension is mainly due to the substantial increase in the proportion of standard deviation. It is also noted that the mean value of surge motion fluctuates between 2.70 and 3.00 m, and the Max and Min values are found for the case of EC 6 to be 5.70 m and 0.02 m, respectively.

Due to the fact that the heave motion of the platform has a mean of 0, it has only three statistics, and their absolute values increase with the increasing wave height. These three statistic values of motion for the case of EC6 are the largest due to the largest wave height, which is 0.36 m at the Max, −0.35 m at the Min and 0.14 m at the Std. When the wind speed changes, the variation rules in the absolute values of Max and Min of the heave motion are the same as that of the thrust with the wind speed, while the corresponding variation for the Std value is opposite to that of the thrust. It is also seen that changes in current speed exert limited influence on the heave motion of the LSBT-FOWT-AC structure. Since EC2, EC7, EC8 and EC9 have the same wave height (3.52 m) and current speed (0.81 m/s), the Std values of the heave motion induced are all 0.034 m, and the corresponding Std value for the case of EC10 only increases by 0.002 m when the current speed becomes 2 m/s. In general, according to the results of these environment conditions, the heave motion range of this LSBT-FOWT-AC structure is very small, within 0.5 m, due to its vertical stability.

The LSBT-FOWT-AC structure sets the wind turbine on the upstream side of the platform for the convenience of maintenance. It is also noted from Figure 13c that the absolute values of Std, Max and Min of the platform pitch all increase with the wave height separately from the mean value. The mean values of these ECs fluctuate between −0.94° and −1.05° due to the excellent stability of the structure. Moreover, the absolute values of the three statistics all increase with the increase in wind speed. Additionally, compared

with platform surge and platform heave, the effect of wind speed on the platform pitch rotation is more evident, and changes in current speed have little effect on it.

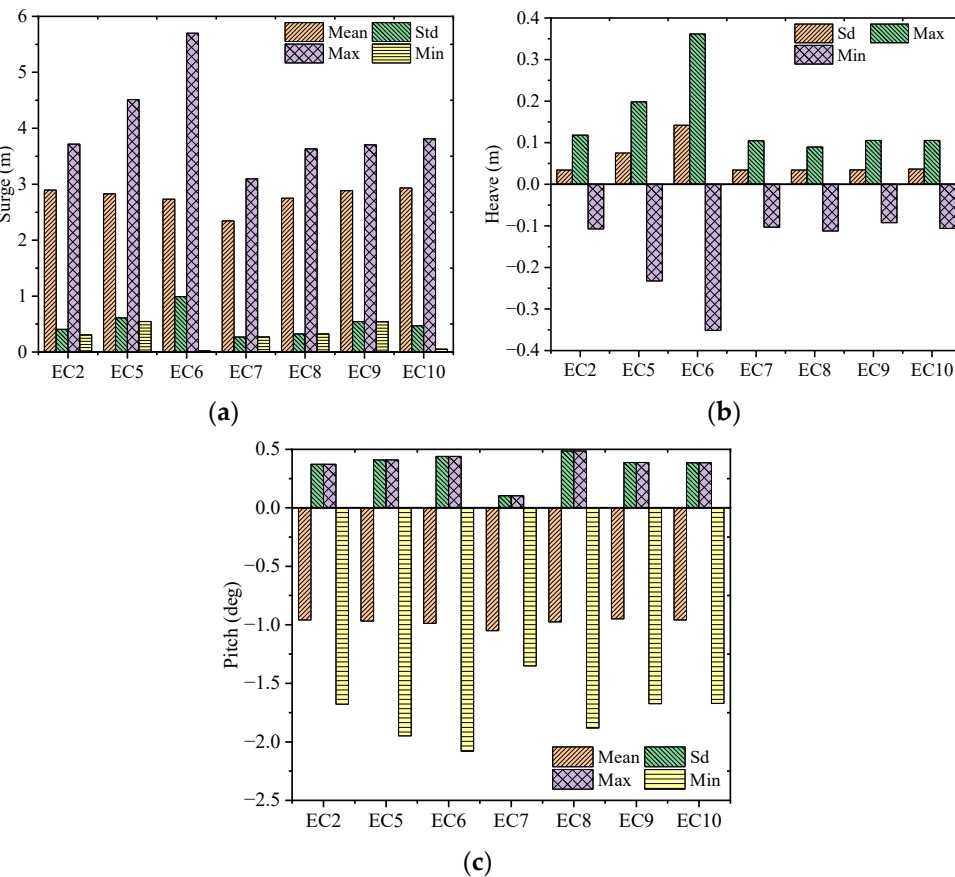

**Figure 13.** Statistics of platform motion response of (**a**) surge, (**b**) heave and (**c**) pitch.

The RootMyc1 is crucial to the overall performance of the integrated LSBT-FOWT-AC system; thus, its statistical value is also of great significance to consider. It can be seen from Figure 14 that the mean value of RootMyc1 fluctuates marginally in these seven ECs, except for the case of EC7 where a smaller mean value of 14,400 kN·m is observed, while all the others fluctuate around 17,000 kN·m. Therefore, the wave height has little effect on the mean value of the RootMyc1, whereas the Max and Min values vary significantly with wave height. It is also observed that the Max and Min values of the RootMyc1 appear in the case of EC8 with the largest wind speed, which are 59,600 kN·m and −27,900 kN·m, respectively. Hence, the influence of wind speed on the RootMyc1 is significant, and the four statistical values all increase with the increase in wind speed. However, the variation of current speed has no effect on the RootMyc1. In general, the RootMyc1 is not sensitive to the change of wave height and current speed, which is mainly affected by the wind.

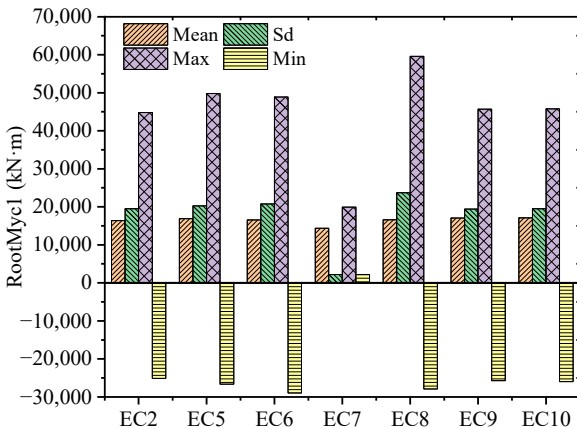

**Figure 14.** Statistics of the RootMyc1.

## 5. Conclusions

In this paper, a novel concept of LSBT-FOWT-AC for 100 m water depth is proposed to support the DTU 10 MW reference wind turbine. Additionally, it is studied by considering wave loads, wind loads and current loads of the environmental conditions in the East China Sea at normal operating and shutdown conditions. The conclusions drawn from the study can be summarized as follows:

(1) The LSBT-FOWT-AC structure is proven to be rather stable by the stability analysis, and its natural periods of platform motion are not consistent with the typical wave frequency and rotor frequency (1P) by free-decay simulation. Therefore, the LSBT-FOWT-AC is well designed in terms of natural period requirements, avoiding structural vibrations excited by first-order wave loads and resonance caused by the 1P effect.

(2) The dynamic analysis results of the platform in the LSBT-FOWT-AC under various sea conditions are studied. Similar to the #1 tension, the surge motion of the platform fluctuates over a wider range as the wind speed and wave height increase. However, the fluctuation range of the platform heave is small, and a coupling effect between the heave and pitch motions is observed. Moreover, the stability of the platform's pitch rotation is excellent, with the maximum pitch angle being within 2° under normal conditions and not exceeding 15° even in extreme sea conditions with a 100-year return period. In summary, the platform of the LSBT-FOWT-AC is greatly affected by wave frequency, and its motion response is within a reasonable range.

(3) From the results of RootMyc1 under various sea conditions, it can be found that when the wind speed is between the rated wind speed and the cut-in wind speed, the order of magnitude for the RootMyc1 is $10^4$. When the wind speed exceeds the rated wind speed, the value of RootMyc1 increases rapidly and the order of magnitude rapidly reaches $10^5$. It has peaks around the wave frequency, rotor frequency (1P), three rotor frequency (3P) and the natural frequency of the first tower mode. Therefore, it is important to avoid these frequency components when designing the LSBT-FOWT-AC structure.

(4) Through the single-variable study of environmental loads, it is noted that wave height has an influence on the Max, Min and Std values of the platform motion; however, it has little effect on the mean value. These four statistics are all sensitive to changes in wind speed. Current speed has little effect on the motion response of the LSBT-FOWT-AC structure compared to wave height and wind speed. Mooring line tension is sensitive to environmental loads such as surge, while RootMyc1 is mainly affected by wind speed. In summary, wind speed has the greatest impact on the dynamic response of the LSBT-FOWT-AC.

## 6. Future Work

In this paper, the hydrodynamic properties and motion responses of the LSBT-FOWT-AC structure were initially investigated by numerical simulations, but there are still many aspects that have not been investigated:

(1) Considering the influence of the existence of the net on the LSBT-FOWT-AC structure;
(2) Considering more complete environmental loads and operating conditions (e.g., fault conditions);
(3) Ensuring that the LSBT-FOWT-AC system meets the design life through fatigue assessment.

**Author Contributions:** Conceptualization, Y.Z., H.Z., X.L. and W.S.; methodology, Y.Z.; software, Y.Z.; validation, Y.Z. and H.Z.; formal analysis, Y.Z.; investigation, Y.Z. and H.Z.; data curation, Y.Z.; writing—original draft preparation, Y.Z.; writing—review and editing, H.Z. and X.L.; supervision, H.Z., X.L. and W.S.; project administration, W.S.; funding acquisition, X.L. All authors have read and agreed to the published version of the manuscript.

**Funding:** This research work was financially supported by the National Natural Science Foundation of China, Grant No. 51939002.

**Institutional Review Board Statement:** Not applicable.

**Informed Consent Statement:** Not applicable.

**Data Availability Statement:** Not applicable.

**Conflicts of Interest:** The authors declare no conflict of interest.

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
