# Peer review of "Design and Dynamic Analysis of a Novel Large-Scale Barge-Type Floating Offshore Wind Turbine with Aquaculture Cage"

_jmse, doi:10.3390/jmse10121926_

Round 1

Reviewer 1 Report

- Page numbering is not following any order.
- The paper has serious english problems, with several parts that require major revision. For instance, in the abstract, "based on the DNV structure, meeting the recommendations of the DNV structure" is probably not what you meant to write.
- Please introduce acronyms/abbreviations before using them, e.g. RootMyc1 in your abstract
- The introduction is confusing. For instance:
    - What do you mean by transition to inshore right after talking about offshore wind? Do you mean shallow waters?
    - In line 97, which "aero-hydro-servo-elastic code"?
    - In "deep-sea aquaculture is also a direction for the development of marine green energy.", what is the link between aquaculture and green energy? Do you mean sustainable fish farming?
- In line 152, what do you mean by "a high degree of freedom"?
- In table 3, which is the diameter of the diagonal column? Because it appears twice in the table. In any case, it would be necessary to illustrate in the figure what you mean by "side column" and "diagonal column", because it is not evident by the text only.
- What do you mean by "when the angle changes from down-flooding shall to the same limiting"? Please rephrase the sentence beginning in line 282 because it is somewhat confusing.
- Please state somewhere in the text the magnitude of the heeling moment, because it is difficult to infer from Figure 5.
- In line 307, I think you meant Table 6 instead of Figure 7.
- Line 327: The heave natural frequency is actually within the typical wave frequencies
- The graphs are of bad quality. Please improve them. Use the available space to make the results easier to see.
- Are you considering second-order hydrodynamics in your radiation/diffraction solution?

Author Response

Dear Reviewer,

Please see the attachment for the detailed reply, thanks.

Best regards,

Zhao Haisheng

Reviewer 2 Report

The reviewer would appreciate the author's efforts and motivation for making this manuscript. However, considerable amount of results have been presented in this work which needs strong scientific knowledge and background, but it still lacks fundamental indicator of a scientific work: 
- Eneglish language needs major improvements, for example in many places very long sentences can be broken down into smaller sentences. 

- Conventional terminologies related to this field should be used, e.g. : In the abstract instead of DNV structure it is suggested to use, DNV rules and standards, 

- in line 16, instead of greatly it is suggested to use significantly or considerable 

- Its not convenient to use a variable in abstract since the audience can not be familiar with the nomenclature before he read whole paper, so instead of RootMyc1 please refer to blade root bending moment 

- the first sentences in the introduction is very in-formal language: It is suggested to use : Floating offshore wind turbines have been proven as one of the main elements of the future green transition in the word.  can begin. 

- in a few places author started a sentence by proposition And which is not a conventional English language 

 - Since the authors have used the proven and conventional numerical models and existing simulation package SESAM, there is no value to present the mathematical theory of the method since this has been extensively explained in the public scientific resources. Author can just cite the method that the analysis is based on. 

- in sec 2.2 author mentioned that the tower shadow effect has not been accounted for which is one of the main contribution in the 1P rotor effect, author may explain what is the source of the 1P effect in the simulation. 

- in equation 11 author shows the Kernel function which does not include the frequency-dependent added mass term, author may clarify the basis for this and why added mass has not been accounted for 

- in Table 1 : please check theblade mass , is it refer to one blade or accumulated mass of three blades 

- in Table 2: second column needs adjustment of the value 

- Sec 3.2: DNV-OS-C301 has several intact stability design checks author has only checked on of the requirements which is not sufficient to prove sufficient intact stability properties other design requirements should be addressed and evaluated 

- The figure 5: The heeling moment looks almost zero , for the intact condition the design wind speed is almost 25 m/s which the wind turbine is in the park position and the heeling moment curve should be calculated accordingly. 

- In the results and discussion: Author has compared the 1P effect with the natural frequency of surge and pitch which is obviously far from each other since for the barge type floater surge motion is a low frequency motion and pitch natural period is above 10 sec for a barge type floater , this comparison does not bring a new knowledge 

- The comparison of the statistics for several response components will not bring a new knowledge for the audience , its better to focus on less number of variable an response statistics which are more important , max and std which are directly proportional to ULS and FLS design checks 

- This work has a good potential to bring valuable results if more scientific approach is taken by the authors 

Author Response

Dear Reviewer,

Please see the attachment for the detailed reply, thanks!

Best regards,

Zhao Haisheng

Reviewer 3 Report

The paper present a novel concept LSBT-FOWT-AC for the 100 m water depth to support the DTU 10 MW reference wind turbine. Authors considered wave loads, wind loads, current loads of the environmental conditions in the China East Sea at normal operating conditions and shutdown conditions for simulating the whole system to obtaine the system response and loads. However, the paper has many mistakes in English, in description of theory. Author may did well in designing, modeling by commercial design tools, but seems not to write carefully. I do believe that this concept is a great idea, I recommend to rewrite the paper to upgrade its quality.

I have some comment as hereunder:

1. It's better to use exactly any word of terminology ocean engineering science.

For instant, these sentences:

- "The motion of  LSBT-FOWT-AC  platform is greatly affected by the wave frequency, and its motion response is within a reasonable range even under the extreme sea conditions of the 100-year return period." 

- "wind speed exerts more significant impact on the dynamic response compared with the wave height and current speed". Don't be confused between wind speed and wind force, wave height and wave force, current speed and current drag force.

- "pitch direction" should be changed to pitch rotation, it is not a direction

- Author should not use this parameter "RootMyc1" without any description at the abstract. 

2. Theoretical methods, At Line 214 velocity potentials, which order did you use in your modeling?

- Did Author considered about drag force from fish net?

3. At Table 2. Main parameters of the barge-type platform

- I confused that how many moonpool to be arranged in the platform? Which moonpool does the Author mention about with dimensions are 20x30x17 (m)?

4. How did Authors consider the effect of fluid inside the moonpool to the platform motions?

5. At the results 

- Could Author explain more why is the rated pitch angle smaller than zero degree? and it seems that all ECs have the same mean pitch value.

Could Authors show more information related to minimum, mean, maximum of surge, heave pitch and including more about naccelle acceleration in the paper? 

Author Response

(The authors gave the same response as above.)

Round 2

Reviewer 1 Report

Thanks for making the requested modifications.

Reviewer 3 Report

Reviewer agree with this version of the manuscript.